# Dependency-Oriented Versus Autonomy-Oriented Help: Inferred Motivations and Intergroup Perceptions

**DOI:** 10.3390/bs14111000

**Published:** 2024-10-28

**Authors:** Huiyue Shi, Yan Dai, Jinzhe Zhao, Liying Jiao, Yan Xu

**Affiliations:** 1Beijing Key Laboratory of Applied Experimental Psychology, Faculty of Psychology and Beijing Normal University, Beijing 100875, China; 201411061109@mail.bnu.edu.cn (H.S.);; 2Department of Psychology, School of Humanities and Social Sciences, Beijing Forestry University, Beijing 100107, China

**Keywords:** dependency-oriented help, autonomy-oriented help, warmth, competence, motive inferences

## Abstract

Intergroup helping is a crucial interaction behavior between groups, which can be classified as either dependency-oriented or autonomy-oriented help. The widely recognized model of intergroup helping relations as status relations proposes that dependency-oriented help enables the helper group to maintain its dominant position. In other words, when a helper group has the motivation to preservation of their intergroup status, it will be more inclined to provide the recipient with dependency-oriented help. However, little research to date has focused on whether members of recipient groups recognize this status maintenance motivation, and how this might influence their inference of altruistic motivations or their perceptions of the helper’s warmth and competence. The results of three experiments involving a total of 677 participants indicated that compared to those receiving autonomy-oriented help, those receiving dependency-oriented help inferred a higher level of status maintenance and a lower level of altruistic motivation of the helper. Furthermore, they also perceived the helpers as having lower levels of warmth and competence. In response, these perceptions led to a reduced willingness to accept offers of help or cooperation. This effect was stronger when the help offered was needed more by the recipients, or when the helpers ignored requests for autonomy-oriented help and provided dependency-oriented help instead. This research complements the existing understandings of autonomy- and dependency-oriented help from the recipient’s perspective, while also outlining potential directions for future research.

## 1. Introduction

Intergroup helping is a common intergroup behavior that occurs between different groups, such as gender, ethnicity, social class, or nationality. While the act of helping is generally seen as being positive, existing theories suggest that helping, especially intergroup helping, is in fact driven by a variety of motivations which can include empathy, but can also include impression management or gaining power over the recipient [1,2]. As such, two distinct categories of intergroup helping have emerged in research: non-strategic help, which refers to helping motivated by benevolent intentions, and strategic help, referring to help motivated by the helper’s in-group favoritism [1,2].

Dependency-oriented help is one type of strategic help, while autonomy-oriented help falls under the category of non-strategic help [2]. Dependency-oriented help refers to assistance which involves providing recipients with a complete resolution to their problems, while autonomy-oriented help refers to the provision of resources and support with the intention of empowering help recipients to address their problems autonomously [3]. Strictly speaking, dependency-oriented help can be both altruistically or strategically motivated, however emerging evidence suggests that such assistance can have adverse impacts on intergroup status relations, potentially resulting in the maintenance or exacerbation of intergroup hierarchy [4,5]. Therefore, dependency-oriented help may sometimes serve as a means for the helpers to maintain their higher status in relation to the help recipients. For example, men in male-dominated contexts might offer women dependency-oriented help as a way of reinforcing their dominance [6], while also, in stereotypically feminine contexts (e.g., housework), seeking dependency-oriented help from women as a means of maintaining their inferior status [7].

Numerous studies have investigated the contextual factors or attributes which may influence helper groups’ decisions to provide either dependency- or autonomy-oriented help, which include the stability of social hierarchy [8], social dominance orientation [9], zero-sum beliefs [10], and paternalistic beliefs [11]. The commonality among these factors is that they are all related to the maintenance of intergroup hierarchy or the maintenance of dominant positions. However, do recipient groups in these situations understand the motivations behind different types of help offered? Though no direct evidence has yet been reported, certain studies have indicated that lower-status groups tend to view help (no specific type) offered by higher-status groups is done so with the intent of maintaining their current dominant position [12]. Furthermore, helping may also even result in a recipient group having a worse impression of the helper group versus no-helping condition when the intergroup trust was low [13]. Meanwhile, one study on interpersonal helping revealed that recipients reported an increase in support for leadership who provided them autonomy- (vs. dependent-) oriented help, an effect mediated by the recipients having a higher level of trust in the helpers’ benevolent intentions [14]. Understanding how recipient groups will respond to different types of help is crucial. If offers of help will function to enhance intergroup status differences, the identification of such a motivation can help weaken entrenched intergroup relations. Furthermore, if those providing help understand the implications of their actions, they may be better able to determine what kind of help to provide so as to not impact ongoing intergroup relations negatively. Therefore, the present research adopted the perspective of the recipient group to explore their inferences regarding the motivations behind different types of help and perceptions of the helper group and finally examined the subsequent interactive decisions.

### 1.1. Intergroup Helping Relations as Status Relations

The widely recognized model of intergroup helping relations as status relations suggests that recipient groups are generally more reluctant to accept dependency-oriented help when the status relation between the helper and the recipient is unstable or illegitimate [4,15]. When the existing hierarchy is unstable, the recipient group (usually the lower-status group) has a stronger incentive to enhance its own intergroup status, however, dependency-oriented help will not facilitate their autonomous development [4,15]. This study, however, takes an interactive perspective and proposes a second possible explanation as to why recipients may not accept dependency-oriented help. When the social relationship between helper and recipient is unstable or imbalanced, a helper in a higher-status position is more likely to perceive a heightened threat to their status, leading them to be more likely to provide dependency-oriented help to those in need [10]. As such, in an imbalanced social context, recipients may assume that the dependency-oriented support being offered is motivated by the helper’s own self-interests, causing the recipient to have less faith in the helper’s benevolent intentions and therefore be more inclined to decline such assistance [16].

The following section will delve into the rationale behind why recipient groups may engage in inferring the motives of their counterparts, as well as the specific motivations they focus on.

### 1.2. Motive Inferences

Research on motive inferences offers valuable insights into the motivations people may deduce from different types of intergroup help. Actions are the most superficial form of social information, followed by intentions, motivations, and stable characteristics or personality traits [17,18]. People not only observe the behavior of others, but they also make spontaneous inferences about the motivations behind that behavior, and the actor’s desired outcomes [17,18]. Research has consistently shown that people tend to over-attribute self-interest as a motive for others’ actions [19,20], a tendency which can be exacerbated by status differences. For example, compared to members of the high-status group, Halabi et al. (2016) found that low-status group members (specifically, Israeli Arabs) were more inclined to perceive assistance from high-status group members (Israeli Jews) as being nonbenevolent [12]. Even when status relations between the two groups are not particularly explicit nor distinct, just the act of providing help tends to position the helper group in a higher status and the recipient group in a lower status [21]. Therefore, in the current study, it was assumed that members of the recipient group would be inclined to make inferences about the helpers’ motivations, and their self-interested motives in particular.

Referring to inferred motivations, the motives of interacting counterparts are deduced according to the perceived consequences the opposite party may obtain from the interaction [17,18]. For instance, when Halabi et al. (2016) asked participants to imagine a situation in which Jewish Israelis offered help to Arab Israelis, the Arab participants perceived the help as a way to achieve domination and reinforce Arab dependency more than the Jewish participants did [12]. In the current study, a key component of inferred motivation was expected to be the potential result of dependency-oriented help, specifically the maintenance of intergroup relative status. We then hypothesized that, compared to autonomy-oriented help, when provided with dependency-oriented help, recipient groups will perceive a higher level of status maintenance motivation. Additionally, as an in-group benefit-oriented motivation, status maintenance motivation will further result in lower perceptions of altruistic motivations. We will discuss in the next section how inferences about status maintenance motives and altruistic motives can influence people’s perceptions of warmth and competence regarding the helper group.

### 1.3. The Effect on Intergroup Perceptions

As previously stated, by inferring the motivations of someone with whom they interact, an individual can then infer their behavioral patterns or characteristics [17]. Warmth and competence, according to the stereotype content model [22], are fundamental aspects that individuals use to form immediate impressions and interpret the behavior of others, and according to the behavior from intergroup affect and stereotypes model, different combinations of warmth and competence elicit different cognitions, emotions, and behaviors [23]. Following this model, the current study also focused on how different types of intergroup help affect recipients’ perceptions of the helper’s warmth and competence.

Perceptions of another’s warmth and competence arise not only from the outcomes of interactive behaviors but also from one’s inference about the other’s motives [24]. If motivations behind helping behaviors are mixed, extrinsic, or self-centered, help recipients will have a negative impression of the helper [24,25,26]. Research has consistently revealed that inferred altruistic motives from a helper positively predict the recipient’s or observer’s perception of a helper’s warmth [24,27]. However, no direct evidence has indicated that there is a relationship between altruistic motives and perceptions of competence. Even so, studies have found that altruistic individuals are often perceived as being of higher status [27,28], which is typically associated with higher perceptions of competence [29,30].

With these understandings in mind, the current study proposed a serial mediation model, as illustrated in Figure 1, that a helper group providing autonomy-oriented help, as opposed to dependency-oriented help, will be perceived by the recipient group as having a lower motivation for status maintenance, implying that they have higher, more altruistic motives, and will subsequently be perceived as having higher warmth and competence by the recipient group.

### 1.4. Overview of the Present Studies

The hypotheses were tested using three studies. Study 1 tested the serial mediation model by manipulating the type of intergroup help provided. The results showed that the mediation model was valid for assessing both warmth and competence perception. Study 2 employed an innovative manipulation to validate the above effects, positing that if the helper group, while providing assistance, simultaneously sought help from the recipient group, then the combined types of help in both directions would interactively influence the recipient group’s perceptions of the helper group. The results showed that participants rated the other party’s warmth and competence lowest when the other party offered dependency-oriented help while requesting autonomy-oriented help. Study 3 was then used to delve deeper, attaining three additional insights. First, when motive inferences were not measured (i.e., when participants were not prompted to consider their perceptions of the helping group), the main effects of the type of help on intergroup perceptions were weakened but remained significant. Second, the effect of the type of help on intergroup perceptions was significant only when the help was of higher importance, such as in the integrated circuit industry, implying that not all forms of assistance are associated with status maintenance. Finally, the recipient group’s perceptions of the helper group’s warmth and competence were positively associated with the recipients’ willingness to accept help and cooperate further.

## 2. Study 1

Study 1 examined the model of autonomy-oriented (vs. dependency-oriented) help → inference of status maintenance motivation → inference of altruistic motivation → perceived warmth/competence. To control for potential intergroup factors (e.g., pre-existing intergroup conflicts), the type of help was manipulated using a virtual country as the helper country and the participants’ own country as the recipient country in the study scenario. Furthermore, considering that different types of help may also influence intergroup perceptions via the perception of relative status rather than inferred motivations, intergroup relative status perception was measured and used as a control variable.

### 2.1. Participants

A total of 120 participants (47 male; *M*_age_ = 30.24; 7.5% high-school graduates, 67.5% college graduates; 25.0% MA or PhD degree-holders) were recruited using Credamo, a participant panel (https://www.credamo.com/), which has over 2.8 million Chinese users spanning all of China’s national and provincial administrative regions. Credamo and its participant panel have gained extensive recognition among social science researchers and have been acknowledged as reliable and credible by authoritative journals [31,32]. Respondents who failed the attention check or the one-item material comprehension question were excluded before data was downloaded for analysis. Ultimately, the sample allowed for the detection of effects as small as *d* = 0.52 with 80% power.

### 2.2. Measures

#### 2.2.1. Helping-Type Manipulation

Two pieces of reading materials were prepared that imitated a journalistic writing style to increase the credibility of the content. Both materials shared the same underlying story which described how, with the assistance of other nations, China had made notable progress in science, technology, and other related areas since the reform and instigation of opening-up policies, which have thereby resulted in significant contributions to China’s national development. The type of help provided was manipulated in the second paragraph. Participants in the autonomy-oriented help group were told that “According to Ministry of Commerce statistics: Since the reform and opening up, Country X has supported China in 162 scientific and technological projects, of which 45% have involved R & D technology licenses, 41% have involved scientific and technological personnel training, 8% have resulted in the output of monopolistic finished products, and 6% have led to the output of critical components (such as chips)”. The dependency-oriented help group material contained different ratios: 8%, 6%, 45%, and 41%, respectively. These statistics were also presented visually in each condition using a pie chart to aid comprehension. To ensure reader attentiveness, a minimum presentation time of 60 s was set for the presentation of the reading material on-screen. Furthermore, an earlier pilot study had been conducted to ensure that the materials successfully manipulated the type of perceived help.

#### 2.2.2. Manipulation Checks

Several questions were used in Study 1 to assess the effectiveness of the manipulation. First, one initial question evaluated participants’ comprehension of the data presented in the reading materials (i.e., “Which two types of scientific and technological support from Country X account for the highest outcome percentages?”). Two later items then focused on the definitions of the two types of intergroup help were used to further evaluate participants’ perception of the nature of the assistance provided by Country X, based on the percentages of the four types of international support. These items were framed as follows: “In general, the support from Country X for our country contributes to enhancing our technological self-reliance”, and “Overall, Country X’s support for our country has deepened our country’s external technological dependence”. Unless otherwise specified, all questions in this paper were scored on a scale ranging from 1 (strongly disagree) to 7 (strongly agree). The Spearman-Brown coefficient for the two items was 0.833.

#### 2.2.3. Inferred Motivation of Status Maintenance

Two items were adapted for Study 1 from Halabi et al. (2016) to assess participants’ inferences regarding Country X’s motive for maintaining its own status (e.g., “To what extent do you think Country X wants to maintain its current international status?”) [12]. All measures of the dependent and mediating variables were assessed using the aforementioned seven-point scale, and Cronbach’s α for this item was 0.88.

#### 2.2.4. Inferred Altruistic Motive

Six items were adapted from Borinca et al. (2021) to evaluate participants’ inference regarding the altruistic motive of Country X (e.g., “Country X takes my country’s developmental needs into account”.) [33]. Three of these items were scored in reverse (e.g., “Country X considers its own national interests more than my country’s interests”.). The Cronbach’s α for inferred altruistic motive was 0.85.

#### 2.2.5. Perceived Warmth and Competence

Twelve items were utilized to assess participants’ perceptions of Country X’s warmth and competence. The warmth perception items used the terms, “warm, friendly, well-intentioned, kind, trustworthy, and sincere”, while the competence perception items used the terms, “competent, efficient, intelligent, skillful, capable, and skillful”. For both measurements, the Cronbach’s α exceeded 0.90.

### 2.3. Procedure

Participants were first asked to read the material about Country X providing assistance to China. They then completed the measurement of variables in the order outlined above. Finally, one item was used to assess participants’ perceived relative status between China and Country X, and demographic variables were collected.

### 2.4. Results

#### 2.4.1. Manipulation Check

Compared to participants in the dependency-oriented group, those in the autonomy-oriented group perceived the assistance from Country X as being more effective in facilitating China’s independence in each specific domain, *t* = 6.43, *p* < 0.001. The result was reversed in terms of the perceived effect of increasing dependency, *t* = −6.01, *p* < 0.001.

#### 2.4.2. The Main Effects of Help Type on Perceived Warmth and Competence

Compared to those in the dependency-oriented help condition (*M* = 3.58, *SD* = 1.68), participants in the autonomy-oriented help condition perceived the helper group to have a significantly higher level of warmth, (*M* = 5.30, *SD* = 1.21), *t*(118) = 6.44, *p* < 0.001, Cohen’s *d* = 1.18. Similarly, participants in the autonomy-oriented help condition also perceived the helper group as being significantly more competent (*M* = 5.91, *SD* = 0.52) compared to those in the dependency-oriented help condition (*M* = 5.22, *SD* = 1.40), *t*(118) = 3.53, *p* < 0.001, Cohen’s *d* = −0.64. Overall, autonomy-oriented help led to a more favorable impression of the helper group among participants than dependency-oriented help.

#### 2.4.3. The Mediating Role of Inferred Motivation

Consistent with our hypothesis, the results showed a significant total effect of help type on warmth perception (*B* = 1.75, *SE* = 0.28, *p* < 0.001, 95% confidence interval (CI) [1.21, 2.30]), among which the total indirect effect (*B* = 1.32, *SE* = 0.21, *p* < 0.001, 95% CI [0.91, 1.74]) accounted for a large part, specifically 75.3%. Results further showed that inferred altruistic motivation mediated the association between help type and warmth perception (*B* = 1.08, *SE* = 0.23, *p* < 0.001, 95% CI [0.64, 1.54]), while the inference of status maintenance motivation did not (*B* = −0.10, *SE* = 0.06, *p* = 0.12, 95% CI [−0.24, −0.00]). The serial mediating effect was also supported (*B* = 0.33, *SE* = 0.14, *p* = 0.02, 95% CI [0.10, 0.65]). After controlling for motivational inferences, the direct effect was only marginally significant (*B* = 0.43, *SE* = 0.23, *p* = 0.06, 95% CI [−0.01, 0.92]).

A significant total effect of help type on perceived competence was also seen (*B* = 0.77, *SE* = 0.20, *p* < 0.001, 95% CI [0.41, 1.17]); 54.5% of this effect was accounted for by the indirect effect (*B* = 0.42, *SE* = 0.16, *p* = 0.007, 95% CI [0.17, 0.77]). Both inferred altruistic motivation (*B* = 0.41, *SE* = 0.15, *p* = 0.007, 95% CI [0.17, 0.75]) and status maintenance motivation (*B* = −0.12, *SE* = 0.06, *p* = 0.05, 95% CI [−0.26, −0.03]) mediated the relationship between help type and perceived competence. It should be noted that the isolated mediating effect of status maintenance motivation was opposite to the overall mediating effect. The serial mediating effect, with a 95% CI of [0.03, 0.29], was marginally significant (*B* = 0.13, *SE* = 0.07, *p* = 0.054). After accounting for motivation factors, the direct impact remained marginally significant (*B* = 0.35, *SE* = 0.18, *p* = 0.05, 95% CI [−0.00, 0.71]). Table 1 displays the regression models, while the route coefficients are shown in Figure 2a,b.

### 2.5. Discussion

Study 1 reveals that, in intergroup helping scenarios, members from the recipient group will perceive a helper country providing autonomy-oriented help (as opposed to dependency-oriented help) as having higher warmth and competence. This was concurrently verified through the serial mediating effects of status maintenance motivation and altruistic motivation. To be noted, the impact of help type was found to be greater on perceived warmth than on perceived competence, while the mediating effect of motivational inferences was also more pronounced for the former. This could be because motivational inferences, influenced by help type primarily, involve altruistic relational motivations, which are more effective in shaping perceived warmth.

Additionally, supplementary analyses treated perceived warmth and competence as dual outcome variables within the same model, allowing for their correlation. This correlation was not seen to affect the primary outcomes, however, a significant positive correlation was revealed between perceived competence and warmth (*r* = 0.39, *p* < 0.001).

## 3. Study 2

In real-world scenarios, intergroup help is often not only given but also sometimes requested. For instance, some countries may offer assistance to other nations while simultaneously seeking international aid from these same countries when confronted with situations such as war, terrorism, urgent political or economic events, or natural disasters. As such, Study 2 aimed to investigate how the pairing of offered and requested help types influences intergroup perceptions, positing that when autonomy-oriented help was provided, the participants’ perception of the helper group would not be significantly influenced by whether the helper requested autonomy- or dependency-oriented help. Conversely, in scenarios where dependency-oriented help was provided, participants would already exhibit heightened motivational vigilance, thus the helper group asking for autonomy-oriented help would further confirm participants’ belief that the helper country’s assistance is intended primarily to reinforce its own interests and status. This would then result in the perceived warmth and competence of the helper group being much lower than in other help-type scenarios.

Study 2 also aimed to reconfirm the main effect and mediating model developed in Study 1. Considering the scarcity of research demonstrating how the requested type of assistance might impact intergroup perceptions, Study 2 also performed an exploratory examination of those effects.

### 3.1. Participants

Using the Credamo participant panel, 495 participants were recruited for Study 2. Considering that both offered and requested help needed to be manipulated in the reading materials for this study, which could lead to confusion, two screening processes were established. Specifically, two forced-choice questions were employed to assess participants’ extraction of the information from the text, in which participants were asked to identify the primary assistance being offered and requested. A total of 69 participants were excluded from data analysis through this step. Subsequently, four items rated using a Likert-style scale were employed to screen out any remaining participants who did not correctly identify the characteristics of the international assistance being provided in their respective reading material (e.g., “Country X’s assistance to our country focused primarily on enhancing scientific and technological capabilities and talent development in the semiconductor industry/addressing the existing shortages of chips and associated finished products”.). If the direction of difference between the two items associated with the two types of help did not align with the help type that was being manipulated, the participant was considered to have misunderstood the nature of the provided help being described in the materials. This step led to the exclusion of a further 113 participants.

After the two-step screening, 313 valid responses remained. Among them, 41.5% were male and 97.8% held a bachelor/associate’s degree or higher. The sample size of each of the four groups ranged from 69 to 89, allowing for the detection of effect sizes of *f* = 0.20 with a power of 94.15%.

### 3.2. Measures

#### 3.2.1. Help Tyle Manipulation

The type of provided and requested help were both manipulated in Study 2. Other than the inclusion of the additional content concerning the chip industry, and the adjustments made to the reported percentage distribution (i.e., 86% vs. 14% or 91% vs. 9%) to emphasize the primary help type, the reading materials were the same those used in Study 1. Regarding the inclusion of requested assistance, the manipulation was articulated as, “Country X has simultaneously requested technical support from our country in high-end technological sectors, including non-ferrous metal smelting, rolling processing, metal products, and specialized equipment manufacturing. Over 90% of their current requests are directed towards enhancing their export of finished and semi-finished products [for the dependency-oriented condition], while the remainder focus on R & D sharing, as well as technology use authorization [for the autonomy-oriented condition]”.

#### 3.2.2. Manipulation Check

In retrospect, the manipulation test questions used in Study 1 (e.g., “Overall, Country X’s support for our country has deepened our country’s external technological dependence”) might have implied to participants that autonomy-oriented help was superior to dependency-oriented help, however, the intention was simply to manipulate the actual help type behaviors, rather than the perceived effectiveness of the help provided. Therefore, more objective phrasing was adopted in Study 2 (e.g., “Country X’s assistance to our country focused primarily on addressing the existing shortages of chips and associated finished products”.). The Spearman-Brown coefficients for the variables of offered help and requested help were 0.953 and 0.944, respectively.

#### 3.2.3. Mediating and Dependent Variables

Consistent with Study 1, perceived altruistic motivation inference, warmth, and competence were again used as the measured variables, with some modifications made in the measurement of inferred status maintenance motivation. In Study 1, the measurement of inferred status maintenance motivation focused primarily on the absolute status of the assisting country, while in Study 2, two statements about relative status were used to broaden the meaning of status maintenance motivation. For example, “Country X is concerned about being surpassed by our country”. The Cronbach’s α for these measures in Study 2 ranged from 0.75 to 0.96.

#### 3.2.4. Controlled Variables and a Potential Alternative Mediator

In addition to gender and perceived relative status, the perceived relative importance of the two help domains was also controlled. The item used to assess this was, “How do you rate the importance of chip-related industries in comparison to non-ferrous metal smelting and other related fields?”.

The perceived equality of the help provided was also considered as potentially serving as a parallel mediator in addition to inferred motivation because unequal giving and requesting could potentially give observers or recipients a negative impression as well. To assess this, participants were asked, “Do you believe that your country and Country X offer each other equal assistance?”.

### 3.3. Procedure

The questions were administered in the following sequence: reading material, manipulation checks, inferred status maintenance and altruistic motivations, perceived warmth and competence, controlled variables, and collection of demographic information. To ensure attentive reading, the reading material and the questions used for the comprehension test questions were each presented individually on separate pages, with a minimum reading time per page set for 20 s.

### 3.4. Results

The dependency-oriented item responses were reverse-coded and averaged with the autonomy-oriented item responses. The results of independent-sample *t*-tests showed that both the manipulation of offered and requested help types were effective (*t*s > 31.69, *p*s < 0.001, mean differences > 3.19).

Regarding the descriptive statistics for the two inferred motivations, as well as the perceptions of warmth and competence, the findings were consistent with expectations, with results shown in Table 2. Participants reported the lowest levels of perceived warmth and inferred altruistic motivation when Country X provided dependency-oriented help but asked for autonomy-oriented help, and rated its status-maintenance motivation the highest in this condition.

ANOVAs were then employed to further examine the statistical significance of the results. The main effect of the provided help type was significant for all four variables, which was consistent with the results of Study 1. The type of requested help only marginally affected inferred altruistic motivation and perceived warmth, while it did not significantly impact the other two variables. None of the four interactions demonstrated statistical significance. Further simple effects analysis found that consistent with our hypothesis, when Country X offered dependency-oriented help, requesting autonomy-oriented help (as opposed to dependency-oriented help) significantly decreased participants’ inference of altruistic motivation, (*F*(1, 306) = 3.95, *p* = 0.048, η^2^ = 0.013, 95% CI of Cohen’s *d*: [−0.645, −0.003]), and perceived warmth, (*F*(1, 306) = 5.10, *p* = 0.025, η^2^ = 0.016, 95% CI of Cohen’s *d*: [−0.690, −0.048]). No difference was observed when the help provided by Country X was autonomy-oriented. More detailed results are presented in Appendix A.

These results and theoretical deductions indicated that the type of requested help would affect perceived warmth only when the provided help was dependency-oriented, through the mediation of inferred altruistic motivation. Therefore, a moderated mediation analysis was conducted which included the equality of the offered help in the model as another mediator. As expected, the results showed that when the offered help was dependency-oriented, requesting autonomy-oriented help led participants to perceive Country X as being significantly less altruistically motivated than if they had requested dependency-oriented help (*B* = −0.35, *SE* = 0.18, *p* = 0.048, 95% CI [−0.699, −0.003]). Accordingly, inferred altruistic motivation mediated the relation between the type of help requested and perceived warmth (mediation effect = −0.27, *SE =* 0.12, *p* = 0.022, 95% CI [−0.492, −0.038]). Furthermore, when provided with dependency-oriented help, Country X requesting autonomy- (vs. dependency-) oriented help also led to greater perceived unfairness (*B* = −1.25, *SE* = 0.22, *p* < 0.001, 95% CI [−1.668, −0.823]), resulting in lower perceived warmth (mediation effect = −0.18, *SE =* 0.08, *p* = 0.018, 95% CI [−0.344, −0.055]). However, when the offered help was autonomy-oriented, none of those effects was significant. Finally, the moderated mediation effect of inferred altruistic motivation did not reach statistical significance (IMM = 0.183, *SE* = 0.190, 95% CI [−0.192, 0.550]). These results are all presented in Appendix A.

### 3.5. Discussion

To further validate the motivational implications of dependency-oriented help, Study 2 transformed the scenario of Study 1 from being a unilateral help transaction to one where both parties requested and were offered help. The results of Study 2 showed that the helper country (Country X) was only perceived to have a significantly lower level of warmth when they provided dependency-oriented help while requesting autonomy-oriented help. This is because when the provided help was dependency-oriented, the request for autonomy- (vs. dependency-) oriented help led participants to infer a lower level of altruistic motivation and fairness, thereby influencing perceived warmth. Notably, these effects did not manifest in either status maintenance motivation or perceived competence. That could be due to the fact that the act of requesting help itself negatively decreases the perceived status of the helper group, thereby weakening their status maintenance motivation and perceived competence, both of which are closely associated with status.

The results of Study 2 regarding the effects of the help type offered also replicated the results of Study 1, as well as the serial mediation effects. The requested help type showed only marginally significant effects on perceived warmth and inferred altruistic motivation, both of which were substantially weaker than the effects of the offered help type. This also indicates that actively seeking help, being provided with help, and being requested for help each have distinct effects on intergroup perceptions.

## 4. Study 3

The aims of Study 3 were threefold. The first was to explore whether the influence of help type on the perceived warmth and competence of the help provider further affected the recipient group’s willingness to accept the assistance offered and cooperate. The second aim was to investigate whether the importance of the help would affect those effects. Regarding this, the importance of the help was considered to be a possible moderator for motivational inferences, in that one might infer the help provider’s motivations in terms of status maintenance only if the actual value of the help offered was important. This would also prove the existence of causal effects in the intergroup helping relationship [34]. The specific hypothesis was that the more crucial the assistance is, the more sensitive the recipient group will be to the motives behind the help being offered. This could potentially result in a paradox whereby the more urgent the help required, the less willing the recipient group will be to accept it. Finally, as there is a chance that the motivational inferences being examined in these studies may not be spontaneous psychological processes but rather induced by the laboratory measurements used, Study 3 did not include the measurement of motivational inferences and the results were compared with those of the previous two studies to determine the impact of the laboratory setting itself.

### 4.1. Participants

Study 3 involved 244 valid participants, recruited using the participant panel Credamo after 59 participants had been excluded based on the screening criteria used in Study 2. Of those remaining, 39.3% of the participants were male, and 96.3% had completed at least a bachelor’s or associate degree. Study 3 employed a 2 × 2 between-subjects design, with group sizes ranging between 48 and 68 participants.

### 4.2. Measures

#### 4.2.1. Manipulation of Help Type and Importance

The manipulation of help type was done following a similar procedure as used in Study 1. In this case, after the reading materials had presented a background introduction, a description of the specific help area was added (i.e., integrated circuit design for the high-importance condition and olive oil for the low-importance condition) to manipulate the help importance. For example, in the high-importance condition, participants were told that:


*…This problem is also reflected in the field of integrated circuit design (ICD, which can be simply be understood as “chips”). ICD is the “brain” of many electronic devices—from small ones such as cell phones, computers, or home appliances, to large ones such as drones, robots, or CNC machine tools. All are inseparable from their ICD. Indeed, it can be said that ICD is the lifeblood of many industries…*


The four specific types of aid were also adjusted according to the respective help area condition.

#### 4.2.2. Manipulation Check

The manipulation test in Study 3 comprised a single reading comprehension question along with two questions checking the importance of manipulation (e.g., “What do you think is the radiating impact of olive oil/ICD on the development of our country?”), and two items assessing the manipulated type of assistance, similar to those used in Study 2. The Spearman-Brown coefficient for the 2-item manipulation check of types of help was 0.713.

All items used in Study 3 were scored using a 7-point scale unless otherwise stated.

#### 4.2.3. Perceived Warmth and Competence

The items used to evaluate perceived warmth and competence were the same as those used in Studies 1 and 2.

#### 4.2.4. Intention to Accept Help and Cooperate Further

The following question was used to assess participants’ willingness to accept the help offered: “As a citizen of our country, do you believe our country should continue to receive assistance in this field from Country X in the future?” A second question was used to assess their intention to cooperate further: “As a citizen of our country, do you believe our government should further its collaboration with Country X?”

#### 4.2.5. Controlled Variables

Perceived relative status was evaluated using the same item as used in Study 1, and controlled for in the statistical model employed in Study 3.

### 4.3. Results

The moderated mediation model was tested using an estimated 5000 bootstraps with 95% CIs. The results showed that the importance of the help moderated both the relationship between help type and perceived warmth (*B*_interaction_ = 0.545, *SE* = 0.271, *p* = 0.046, 95% CI = [0.011, 1.080]), as well as the mediating role of perceived warmth on willingness to accept help (index of moderated mediation = 0.311, *SE* = 0.151, 95% CI = [0.019, 0.607]) and further cooperation (IMM = 0.308, *SE* = 0.169, 95% CI = [0.015, 0.678]). Specifically, when the field of assistance was considered to be more important, the offer of autonomy-oriented (as opposed to dependency-oriented) help led the recipients to perceive a higher level of warmth, therefore making them more willing to accept the help being offered and to cooperate further. However, when the field of assistance was deemed as being unimportant, no differences were seen between the two help types in terms of perceived warmth. Furthermore, neither the interaction effects on perceived competence nor the moderating effects on the mediating role of perceived competence reached statistical significance. These results are illustrated in Figure 3a,b.

Notably, the main effects of help type on both perceived warmth and competence in the high-importance condition were smaller, although still significant (Figure 4), in Study 3 than those found in Study 1. This difference could be attributed to the non-inclusion of the inferred motivation measurements in Study 3, which may have resulted in reduced engagement in motivational inference processing among participants. To test this theory, data from the high-importance condition in Study 3 was used along with data taken from Study 1 to examine whether the presence or absence of a motivation-inference reminder influenced the effects of help type on perceived warmth or competence. The results showed a significant moderating effect of the reminder (*B*_interaction–warmth_ = 0.672, *SE* = 0.327, *p* = 0.041, 95% CI [0.028,1.315]; *B*_interaction–competence_ = 0.471, *SE* = 0.226, *p* = 0.039, 95% CI [0.025, 0.917]), which suggests that the positive effect of autonomy- (vs. dependence-)oriented help on intergroup perceptions relies partly on reminders to infer motivations. However, even without such reminders, the effects nonetheless remained significant.

### 4.4. Discussion

The major discovery of Study 3 was that of the “help-receiving paradox” which suggests that the more important or necessary the assistance, the more likely the recipient group is to refuse that help or cooperate with the helper as they are less likely to believe that the helper is acting with altruistic intentions. Furthermore, the strength of the inferred motivation was shown to be impacted by whether the inferred motivation was in fact assessed during the experiment. Specifically, the effects of the help type on perceived warmth and competence were amplified when status maintenance and altruistic motivation inferences had also been measured. In contrast, when these were not measured as a part of the study, the impact of help types on perceived warmth diminished, and its effect on perceived competence became insignificant. Besides, it is important to note that cooperation implies reciprocity, which is motivated differently from altruistic assistance. Future research examining the willingness to provide help should consider using the willingness to offer altruistic assistance as a substitute for the willingness to cooperate.

## 5. General Discussion

In everyday situations, many people do not always feel appreciation for or even like those that offer them help, especially when the helper’s motivations seem to be strategic. Using three studies, the current research found that, when provided with autonomy-oriented help, people tend to perceive the helper group as being warmer and more competent than when provided with dependency-oriented help. This is because when provided with autonomy- (vs. dependency-) oriented help, members of the recipient group tend to infer that the helper group has a lower level of self-interested motivation and thus a higher level of altruistic motivation. In addition, the importance of help also influences the abovementioned effects. Help type has a greater impact on intergroup perceptions towards the helper group when the help is of high importance. Furthermore, in real-life situations, where different groups may possess strengths in different domains, this can lead to situations where a helper group may also be a help-seeking or recipient group in another domain. Study 2 addressed this situation and found that asking for autonomy-directed help from another group without offering them the same type of help can lead to a deterioration of the cooperative relationship.

Dependency-oriented intergroup helping is a critical way to solidify the relative status of groups, however, this negative effect is often overlooked due to the nice exterior of providing help. Therefore, it is important that recipient groups identify the motivations behind dependency-oriented help, and selectively seek or accept help that will benefit the long-term development of both of their groups, and the relationship between them. The current study confirms that recipient group members did engage in corresponding motivational inference processes, as well as demonstrating subsequent coping styles such as refusing or cooperating with the help offered. This study also provides valuable insights for those offering assistance. For instance, in diplomatic relations, if both parties can infer each other’s psychological processes and behavioral intentions more accurately, they can effectively leverage intergroup support to enhance their relationship. Similarly, leaders can avoid negative perceptions from subordinates by being mindful of how they provide assistance, ensuring that their help is both appropriate and well-intentioned. As another example, if women in the workplace recognize the motivations behind paternalistic help offered by male employees, or if they actively seek help that is more conducive to increasing their independence in male-dominated spheres, they may be able to elevate their status and avoid the solidification of their intergroup status that could result from accepting dependence-oriented help. However, these understandings can be applied to any groups with existing group status discrepancies.

In a broader sense, this research contributes an interactive perspective on our understanding of intergroup helping. In previous studies, providing help has usually been treated as unidirectional behavior wherein helper groups do not pay attention to the responses of recipient groups, and helping decisions are driven primarily by the helper group’s motivations. However, by adopting the recipient’s perspective, this research found that recipients also make inferences about the motivations behind the help offered, and adjust their impressions of the helper group accordingly. To extend these understandings further, it could be that when the recipient group is less willing to accept dependency-oriented help due to its concerns about that being the type of help offered, the helper group may then adjust the type of help it provides. Furthermore, as both parties’ perceptions and behavioral decisions continue throughout their interaction with one another, perhaps this will also eventually lead to adjustments made in the helping behavior. Future research should investigate such interactions in their entirety.

Self-interested motives in strategic intergroup help refer to more than simply motives of status maintenance. For example, some help may be motivated by impression management, meaning that the helper group wants to be perceived by others as being warmer or more competent. These motivations may also reduce the recipient’s perception of warmth by reducing their inference about altruistic motives in helping. However, we posit that the motivation to maintain status remains one of the primary factors that can diminish the perception of warmth. This is because a motive of impression management may be non-altruistic, however it will not cause harm to the help recipients; meanwhile, a motivation of status maintenance will result in some damage to the help recipients by maintaining or lowering the status level in relation to that of the help providers. In this case, the latter is likely to have a much greater negative impact on the recipients’ impression of the helper group.

In addition to inferred motivations, the recipient group’s feelings regarding the help itself can also influence its perceptions of the helper group. For example, in line with the theory of self-determination [35], Carbonneau et al. (2019) found that receiving autonomy-oriented support from one’s partner was positively associated with relationship satisfaction, as it could help the recipient meet their psychological need for autonomy within the relationship; in contrast, directive support was either unrelated or negatively associated with relationship satisfaction [36]. Similarly, Halabi et al. (2021) argued that dependency-oriented help is more psychologically threatening than autonomy-oriented help [16]. When given an opportunity to exert control, participants reported less negative responses towards outgroup helpers and to the outgroup as a whole. This may account for effects that inferred motivation could not explain.

Going forward, three future research directions are proposed. First, researchers can replace the virtual international relations scenario used in the current research with group divisions more familiar to study participants, such as educational background, work role, or gender, to explore the effects of variables such as group status differences, competitive relationships, and the stability of the status relation on the studied effects. An individual’s own identification with a group can also be a potential influencer. For example, one study found that only those female participants who identified strongly with feminists showed a significant preference for autonomy-oriented help from men [37]. Second, although dependency-oriented help can trigger the inference of relevant self-interested motivations, whether the effect of this on not accepting the provided help is positive or negative requires further investigation. Third, as this study focuses solely on intergroup assistance within the economic sector, future research could examine the effects in other contexts of intergroup help, such as politics and military frameworks. The varying impacts of intergroup assistance on relative status in these areas may lead to diverse motivation inferences, which could, in turn, influence intergroup perceptions in unique ways. Finally, as noted previously, future research should examine an entire interaction pathway to explore whether and how helper groups might further adjust their helping strategies in response to recipient groups’ potential negative attitudes towards them.

In conclusion, the current study found that being offered autonomy-oriented help, as opposed to dependency-oriented help, leads the recipient group to perceive the helper group as having more warmth and competence. This is partially explained by the spontaneous inferences made regarding status maintenance motivation and altruistic motivations, and can further influence a recipient group’s willingness to accept the help being offered and to cooperate with the other party. Furthermore, the more crucial the help, the greater the recipient group’s preference for autonomy-oriented help.

## Figures and Tables

**Figure 1 behavsci-14-01000-f001:**
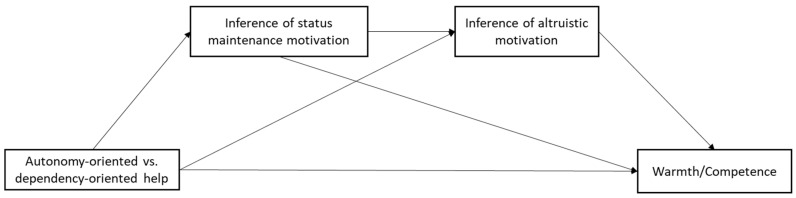
The serial mediation model type of helps on intergroup perceptions.

**Figure 2 behavsci-14-01000-f002:**
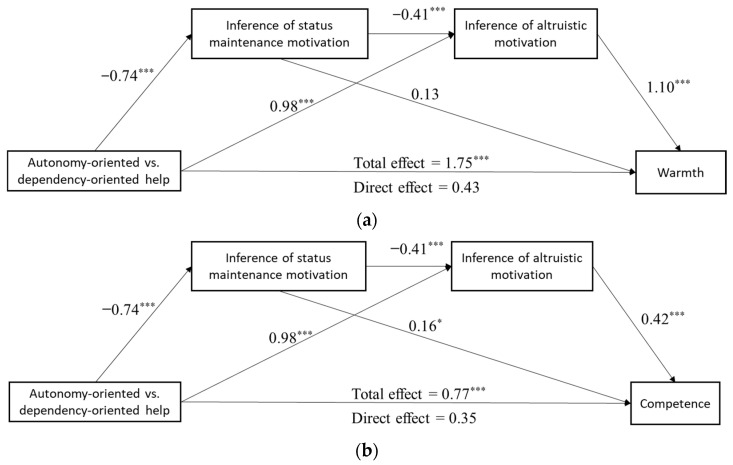
(**a**) The mediating model of inference of motives. (**b**) The mediating model of inference of motives. Note. * *p* < 0.05. *** *p* < 0.001.

**Figure 3 behavsci-14-01000-f003:**
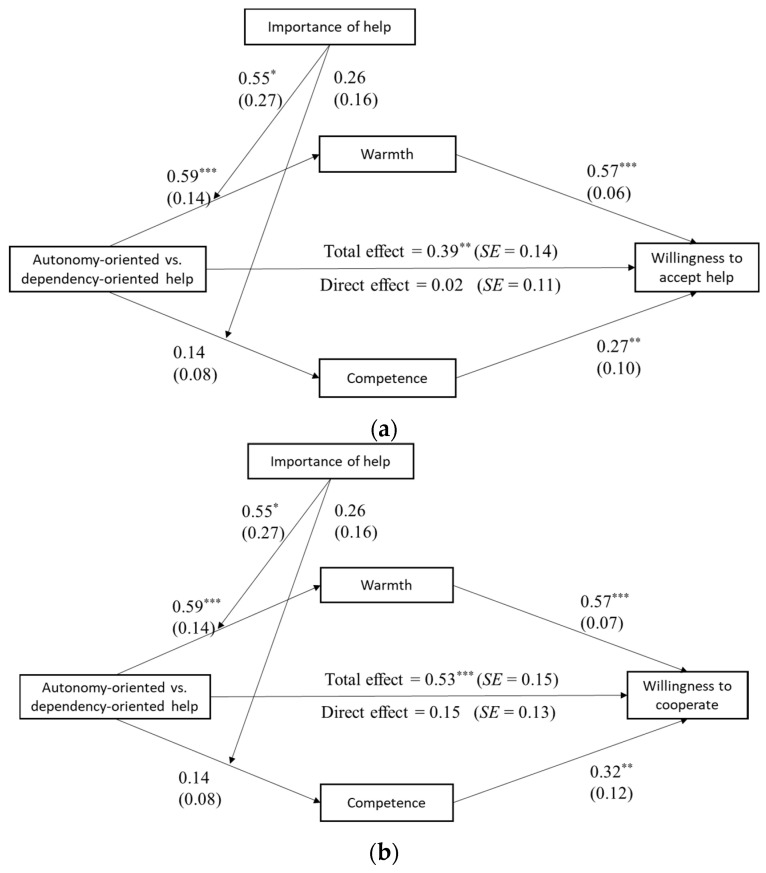
(**a**) The moderated mediation model of the willingness to accept help. (**b**) The moderated mediation model of willingness to further cooperate. Note. * *p* < 0.05. ** *p* < 0.01. *** *p* < 0.001.

**Figure 4 behavsci-14-01000-f004:**
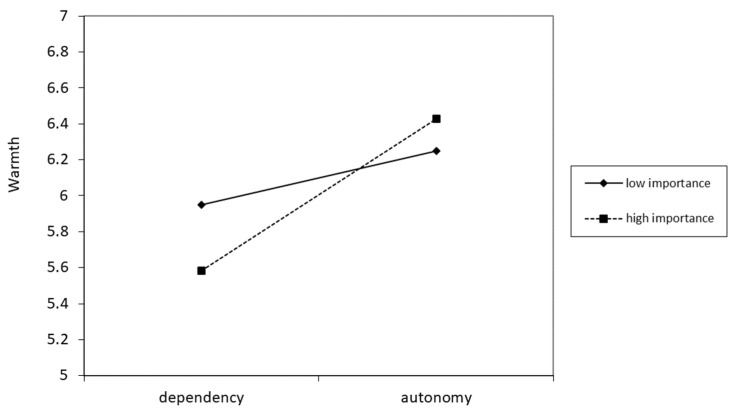
The moderating effect of help is importance in the type of help on warmth.

**Table 1 behavsci-14-01000-t001:** Regression results.

	Warmth	Status Maintenance	Altruistic	Warmth
(Intercept)	3.202 *** (0.926)	5.514 *** (0.817)	5.533 *** (0.685)	−1.235 (0.890)
Gender	0.152 (0.278)	0.624 * (0.246)	0.175 (0.183)	0.160 (0.190)
Relative status	0.056 (0.156)	0.091 (0.138)	–0.036 (0.100)	0.124 (0.104)
Type of help	1.747 *** (0.274)	–0.740 ** (0.241)	0.979 *** (0.182)	0.432 (0.211)
Status maintenance motive			–0.409 *** (0.067)	0.129 (0.080)
Altruistic motive				1.100 *** (0.097)
*R* ^2^	0.262	0.137	0.455	0.682
Adj. *R*^2^	0.243	0.115	0.436	0.668
	**Competence**	**Status Maintenance**	**Altruistic**	**Competence**
(Intercept)	3.653 *** (0.655)	5.142 *** (0.817)	5.533 *** (0.685)	1.402 (0.888)
Gender	0.181 (0.197)	0.624 * (0.246)	0.175 (0.183)	0.116 (0.190)
Relative status	0.264 * (0.111)	0.091 (0.138)	–0.036 (0.100)	0.280 ** (0.104)
Type of help	0.769 *** (0.193)	–0.740 ** (0.241)	0.979 *** (0.182)	0.351 (0.211)
Status maintenance motive			–0.409 *** (0.067)	0.159 * (0.080)
Altruistic motive				0.418 *** (0.097)
*R* ^2^	0.141	0.137	0.455	0.262
Adj. *R*^2^	0.119	0.115	0.436	0.230

Note. Unstandardized regression coefficients are displayed, with standard errors in parentheses. * *p* < 0.05. ** *p* < 0.01. *** *p* < 0.001.

**Table 2 behavsci-14-01000-t002:** Descriptive Statistics for 2 × 2 Grouping.

Offer	Request	*N*	Warmth	Competence	Altruistic	State Maintenance
Autonomy	Autonomy	89	5.25 (1.07)	5.88 (0.81)	4.06 (1.24)	4.89 (1.20)
Autonomy	Dependency	69	5.17 (1.25)	5.86 (0.57)	4.15 (1.15)	4.82 (1.21)
Dependency	Autonomy	69	3.72 (1.54)	5.45 (0.85)	2.88 (1.15)	5.73 (0.73)
Dependency	Dependency	86	4.30 (1.34)	5.50 (0.88)	3.26 (0.93)	5.57 (0.77)

Note. M (SD) denotes Mean and Standard Deviation.

## Data Availability

The data presented in this study are available on request from the corresponding author.

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
