# Peer review of "Dependency-Oriented Versus Autonomy-Oriented Help: Inferred Motivations and Intergroup Perceptions"

_behavsci, 2024, doi:10.3390/bs14111000_

Round 1

Reviewer 1 Report

Comments and Suggestions for Authors

This article presents and interesting study on intergropup relations and motivational inferences about autonomy or oriented help. However there are some issues that can be clarify.

Author Response

Comment 1: However, there are some issues to be referred to more precisely. On line 237, the authors mention internal consistency with two items. Is this adequate and is there a minimum number of items for this analysis? On line 264, there is a Cohen's value that is confusing or that the authors should explain (d=1.18).

Response 1: 

Thank you for your professional statistical advice! We have reviewed some literature and found that it is indeed inappropriate to calculate Cronbach's α coefficient for only two items, as this coefficient is primarily used to assess the internal consistency among multiple items (typically three or more).

Based on the comparison results of various 2-item scale reliability indicators by Eisinga et al. (2013), we have chosen to adopt the Spearman-Brown statistic as our reliability indicator. The updated reliability indicator has been incorporated into the revised manuscript. In Study 1, the Spearman-Brown coefficient for the 2-item manipulation check of types of help was 0.833. In Study 2, the Spearman-Brown coefficients for the variables of offered help and requested help were 0.953 and 0.944, respectively. In Study 3, the Spearman-Brown coefficient for the 2-item manipulation check of types of help was 0.713.

We also recalculated the Cohen's d in line 264. The result shows that the value is correct: Cohen's d = (5.3 - 3.58) ⁄ 1.463984 = 1.174876. However, due to our previous oversight, we did not include a negative sign. We sincerely apologize for this mistake and thanks a lot for your careful review.

Reference: Eisinga, R., te Grotenhuis, M., & Pelzer, B. (2013). The reliability of a two-item scale: Pearson, Cronbach, or Spearman-Brown? International Journal of Public Health, 58(4), 637–642. https://doi.org/10.1007/s00038-012-0416-3

Comment 2: Another issue is related to the distribution of the sample and its sociodemographic characteristics. In studies 1 and 3, the majority of the sample are women. There is no reference on the different motivations perceived by men compared to women. However, there is a vast literature on differences between men and women in volunteering activities, stereotypes or causal attributions about poverty or perceptions in vulnerability groups. In addition, the participants show a higher educational level, and the authors do not refer to other ideological characteristics, knowledge of their volunteer activities or, for example, their involvement or not in non-governmental organizations. All these characteristics may modulate the perception and inferred motivations of perceived help by other groups.

Response 2: 

Your suggestion is a particularly helpful reminder, and we would like to clarify a few points regarding this. First, we did not impose any restrictions or guidance on gender during the questionnaire collection process. Second, aside from the finding in Study 1 that gender exhibited a predictive effect on the inference of status maintenance motivation (see Table 1a or 1b), no other significant effects were observed for the remaining variables and studies. These results are presented in Tables 1a and 1b, as well as in Tables 1, 3, and 4 of the supplementary materials. For this reason, we did not specifically focus on the issue of gender. However, we strongly agree with your point that it is important to consider the influence of gender in motivation inference. Lastly, the absence of average sample attributes and the lack of collected demographic information are limitations of the current study. We reflect on this and think it results from our failure to diversify our data collection approach. We greatly appreciate your emphasis on sample representativeness! Because this not only encourages our team to reflect on the current research but also serves as a strong reminder for us to plan our samples more thoughtfully before conducting future psychological studies.

Comment 3: Finally, as for the Discussion sections of each study, they can be integrated into the final discussion and the structure of the article can be lighter.

Response 3: We have carefully discussed and reviewed the discussions for Studies 1 to 3 to eliminate any redundant expressions. We have removed the following paragraphs to enhance the conciseness of the manuscript:

  • These results also suggest that altruistic motivation plays a more significant role as a mediating variable than status maintenance motivation. This is because status maintenance motivation alone cannot mediate the relationship between help type and perceived warmth. Meanwhile, status maintenance motivation was seen to have an opposite effect on the total indirect effect in terms of predicting perceived competence. This could be because the motivation to maintain status could lead participants to perceive the helper group as possessing greater agency, and agency is typically strongly linked with competence (Carrier et al., 2014).
  • While both providing and being asked for help simultaneously may be uncommon in everyday life in a group context, mutual assistance between individuals can also sometimes be accompanied by the activation of group identities. For instance, a woman can receive dependency-oriented help in a male-dominated domain while simultaneously being requested for autonomy-oriented help in a female-dominated domain. When such identities are activated, inferences regarding motivations related to intergroup hierarchies are likely to emerge.

Comment 4: It has been important to use the applicability of the results to intergroup relations between men and women. However, it may also be interesting to use other examples of the applicability of the results in inter-country contexts, whose relationships are defined by inequality of resources, power and development. Delving into dependency-autonomy oriented aid and the reactions and policy responses in this context between countries can enrich the discussion paper.

Response 4:  We wholeheartedly agree with your suggestions regarding the practical applications of our research. Therefore, we have added examples of application scenarios between countries and between superiors and subordinates in the general discussion to enhance the real-world significance of our study. Thank you once again for recognizing the practical implications of our research.

Reviewer 2 Report

Comments and Suggestions for Authors

Dear all,

I particularly appreciate your efforts in writing this manuscript. I have therefore carefully reviewed this paper and would like to offer my feedback and comments.

The manuscript offers valuable insights into how different types of help are perceived by recipient groups and how inferred motivations affect intergroup relations.

Overall, the manuscript is well-written, methodologically sound, and makes a meaningful contribution to intergroup helping literature by investigating the recipient's perspective on dependency- and autonomy-oriented help. In fact, the research question is well defined, as is the analysis of the gaps in the current literature in this field and the innovations this manuscript can offer. Indeed, the literature review appears comprehensive and effectively places this work in the broader academic context, providing new insights into the field of intergroup helping. The research methodology employed is in line with the stated aims and the results are clearly presented. Finally, the conclusions drawn are supported by the information presented throughout the paper.

Despite these strengths, I would like to recommend some modifications and implementations.

-            The Introduction section could benefit from a clearer style to guide the reader through the theoretical rationale and research gaps. For example, the transition from intergroup status models to motivational inferences is sudden and could be smoother.

-            The tables are clearly formatted, but small changes can be made to improve readability. For example, in Table 2 (p. 10), it might be useful to include additional labelling (e.g. clarifying the position of Mean and Standard Deviation). The presentation of the data is good overall, but small adjustments can make the results more accessible to readers.

-            I find the potential for application of this work very promising. Therefore, although the discussion hints at the implications of the results, more in-depth reflection on the practical applications would be very useful. The results of the study might have implications for international diplomacy or organisational behaviour, which are only briefly mentioned.

In conclusion, the manuscript as a whole presents a valuable contribution to the field. I hope that the adoption of the recommended improvements will be helpful in enriching the present work.

Reviewer 3 Report

Comments and Suggestions for Authors

The article addresses the dynamics of intergroup helping by presenting three studies. According to the first study's results, the recipient evaluates the helper more positively when they provide autonomy-oriented help rather than dependency-oriented help, and altruistic motivation has a stronger moderating effect than status-maintaining motivation. The results of the second study indicate that when we ask for autonomy-oriented help and instead receive dependency-oriented help, we evaluate the helper more negatively. The third study highlights how willing we are to accept help depending on whether the help is dependency-oriented or autonomy-oriented, and how crucial the help is to our situation. The theoretical background is detailed and fits well with the research objectives. The presentation of the studies is adequate; however, according to the reviewer, a more detailed presentation of the stimulus material would be justified, as the difference between autonomy-oriented and dependency-oriented help is not clear to the reader. A general critical observation regarding the study is that the author frequently refers to everyday interpersonal or intergroup situations in the discussion, where power relations are an integral part of the dynamics of groups or interpersonal interactions. Meanwhile, in the methodology used in the study, the power relations have been systematically excluded as a variable. This raises the question of how applicable the results are to real-life situations. Based on the manipulation used, which portrays transactions in an economic framework, it also raises the question of to what extent this can be considered help and to what extent it cannot. In the context of foreign policy business, can we talk about “help”, which the author measures in the study using the following item: “As a citizen of our country, do you believe our government should further its collaboration with Country X?" Collaboration implies reciprocity, so it raises the question of whether altruistic help even can be considered in this context. It would be worth reflecting on these critical points in the General Discussion section. Beyond these critical remarks, the overall impression is that a very dense and exciting research material has been presented, for which I congratulate the authors.
